# Dendrogenin A Synergizes with Cytarabine to Kill Acute Myeloid Leukemia Cells In Vitro and In Vivo

**DOI:** 10.3390/cancers12071725

**Published:** 2020-06-29

**Authors:** Nizar Serhan, Pierre-Luc Mouchel, Philippe de Medina, Gregory Segala, Aurélie Mougel, Estelle Saland, Arnaud Rives, Antonin Lamaziere, Gaëtan Despres, Jean-Emmanuel Sarry, Clément Larrue, François Vergez, Laetitia Largeaud, Michel Record, Christian Récher, Sandrine Silvente-Poirot, Marc Poirot

**Affiliations:** 1Unité Mixte de Recherche (UMR) 1037, Cancer Research Center of Toulouse (CRCT), Institut National de la Santé et de la Recherche Médicale (INSERM) Université de Toulouse, Team Cholesterol Metabolism and Therapeutic Innovations, Equipe labellisée par la Ligue Contre le Cancer, 31037 Toulouse, France; nizar.serhan@inserm.fr (N.S.); philippe.de-medina@inserm.fr (P.d.M.); gregory.segala@hotmail.fr (G.S.); aurelie.mougel@yahoo.fr (A.M.); michel.record@inserm.fr (M.R.); 2Cancer Research Center of Toulouse (CRCT), Unité Mixte de Recherche (UMR) 1037 Inserm/Université Toulouse III-Paul Sabatier, ERL5294 Centre national de la recherche scientifique (CNRS), Team Drug Resistance and Oncometabolism in Acute Myeloid Leukemia, 31037 Toulouse, France; pierre-luc.mouchel@inserm.fr (P.-L.M.); estelle.saland@inserm.fr (E.S.); jean-emmanuel.sarry@inserm.fr (J.-E.S.); Clement.Larrue@unige.ch (C.L.); 3Service d’Hématologie, Institut Universitaire du Cancer de Toulouse-Oncopole, CHU de Toulouse, Université de Toulouse, 31400 Toulouse, France; vergez.francois@iuct-oncopole.fr (F.V.); largeaud.laetitia@iuct-oncopole.fr (L.L.); 4AFFICHEM, 31400 Toulouse, France; a.rives@dendrogenix.com; 5Dendrogenix, 4000 Liège, Belgium; 6Laboratory of Mass Spectrometry, Institut National de la Santé et de la Recherche Médicale (INSERM) ERL 1157, Centre national de la recherche scientifique (CNRS) Unité Mixte de Recherche (UMR) 7203 LBM, Sorbonne Universités-UPMC, CHU Saint-Antoine, 75012 Paris, France; antonin.lamaziere@upmc.fr (A.L.); gaetan.despres@upmc.fr (G.D.)

**Keywords:** Acute myeloid leukemia, Dendrogenin A, tumor suppressor, cholesterol metabolism, synergy, primary cancer cells

## Abstract

Dendrogenin A (DDA) is a mammalian cholesterol metabolite that displays potent antitumor properties on acute myeloid leukemia (AML). DDA triggers lethal autophagy in cancer cells through a biased activation of the oxysterol receptor LXRβ, and the inhibition of a sterol isomerase. We hypothesize that DDA could potentiate the activity of an anticancer drug acting through a different molecular mechanism, and conducted in vitro and in vivo combination tests on AML cell lines and patient primary tumors. We report here results from tests combining DDA with antimetabolite cytarabine (Ara-C), one of the main drugs used for AML treatment worldwide. We demonstrated that DDA potentiated and sensitized AML cells, including primary patient samples, to Ara-C in vitro and in vivo. Mechanistic studies revealed that this sensitization was LXRβ-dependent and was due to the activation of lethal autophagy. This study demonstrates a positive in vitro and in vivo interaction between DDA and Ara-C, and supports the clinical evaluation of DDA in combination with Ara-C for the treatment of AML.

## 1. Introduction

Dendrogenin A (DDA) is a natural onco-suppressor metabolite at the crossroad between the cholesterol and histamine metabolism [1,2,3,4]. DDA was first imagined and conceived as a conjugation product of cholesterol-5,6-epoxide when the cholesterol-5,6-epoxide hydrolase (ChEH) was identified and characterized as a pharmacological target, tamoxifen, a drug widely used in the hormonotherapy of breast cancers [5,6]. ChEH subunits: the 3β-hydroxysteroid-Δ7,Δ8-isomerase (EBP or D8D7I) and the 3β-hydroxysteroid-Δ7-reductase (DHCR7) are known to control developmental programs [7], it was therefore postulated that cholesterol-5,6-epoxide could be metabolized into compounds with cell differentiation properties. Several conjugation products of 5,6-EC with natural amines known to concentrate at the proximity of ChEH were synthesized, and amongst these, DDA emerged as a compound that induced the re-differentiation of cancer cells of various tissue origins into cells with phenotypic characteristics of normal-like cells [1,4,8,9,10]. It was shown that DDA was a metabolite present in various tissues including blood [4]. This pioneering study also established that a deregulation of the DDA metabolism occurred in cancers with a decreased DDA level, below pharmacologically active concentrations in tumors and cancer cells compared to normal cells and healthy tissues [4,11]. In addition, to trigger cancer cell re-differentiation, longer exposure times or µM concentrations of DDA induced cell death in cancer cells of various tissue origins [1,8,10], including acute myeloid leukemia (AML) cell lines and primary tumors of AML patients in vitro and in vivo [1,10]. The anticancer properties of DDA were found to be independent of the cytogenetic and molecular classification of AML [1]. The progenitor/leukemia stem cell subpopulation was found to be sensitive to DDA as was the bulk cancer cell population [1], highlighting a major difference with conventional chemotherapeutic agents such as cytosine arabinoside (cytarabine or antimetabolite cytarabine (Ara-C)). Interestingly, DDA decreased AML cells in bone marrow and brain, and reduced the total cell tumor burden in bone marrow and spleen in established disease models (e.g., orthotopical engraftment of HL-60 cells and three primary AML patient cells injected into NOD/SCID/IL2Rγc-deficient mice) [1].

It was reported that DDA kills AML cells through the induction of lethal autophagy [1,12,13,14]. This cytotoxic effect was mediated by the biased activation of the nuclear receptor LXRβ (NR1H2), and the inhibition of ChEH and its D8D7I subunit involved in cholesterol neo-synthesis [1]. DDA stimulates, in an LXRβ-dependent manner, the expression of genes encoding NR4A1 and NR4A3 transcription factors, lysosomes and autophagy biogenesis regulators (TFEB and LC3) [1]. Importantly, NR4A1 and NR4A3 were shown to be AML tumor suppressors [15,16]. On the other hand, DDA induces the accumulation of delta-8 sterols, such as zymostenol and 8-dehydrocholesterol though D8D7I inhibition; these sterols are known to contribute to lysosome and autolysosome formation in cancer cells [12,17,18].

Ara-C remains, with anthracyclines, the gold standard for AML treatment, despite a poor clinical outcome, especially in elderly patients [19,20,21]. Ara-C is a pyrimidine nucleoside analog, which inhibits DNA biosynthesis and kills AML cells through a caspase-dependent apoptotic process [22]. In addition, Ara-C triggered a protective autophagy, which may impair its anticancer efficacy [23], and constitutes a mechanism of resistance to treatment. This protective autophagy has been reported for numerous anticancer drugs, as limiting their cytotoxic efficacy [24,25]. These observations led to therapeutic strategies aiming to inhibit protective autophagy processes in several cancers [25]. On the other hand, recent observations suggested that autophagy remained an important mechanism required for the activation of an anticancer immune response, which is very important for long term survival in patients [26], showing that maintaining autophagy during chemotherapy would be profitable. Thus, it would be of interest to evaluate drugs such as DDA that could activate a dominant autophagic cell death to counteract protective autophagy.

Since combination therapies were proposed earlier as an attractive alternative to improve the therapeutic outcome in patients [27], we have conducted tests to determine whether DDA could potentiate Ara-C cytotoxicity and sensitize AML cells to Ara-C.

## 2. Materials and Methods

### 2.1. Materials

Chemicals and solvents were from Sigma-Aldrich (Merck KGaA, Darmstadt, Germany) unless otherwise specified. A list of the primary antibodies is provided in the Appendix A. Secondary antibodies labeled with horseradish peroxidase were purchased from Promega. The caspase inhibitor z-VAD-fmk was from Calbiochem (Merck KGaA, Darmstadt, Germany) and the cyto-ID autophagy detection kit was purchased from Enzo Life Science (Villeurbane, France). DDA was synthesized as previously described [1], and its purity was confirmed as greater than 99% by liquid chromatography/ mass spectrometry (LC/MS).

### 2.2. Cell Lines and AML Primary Samples

Human myeloid leukemia cell lines HL-60 (CCL-240), KG1 (CCL-246) and MV4-11 (CRL-9591) were purchased from the American tissue culture collection (ATCC). Cell lines were grown in RPMI-1640 medium with Glutamax (Lonza, Basel, Switzerland), supplemented with 10% fetal bovine serum (Dominique Dutsher, Brumath, France) and 100 units/mL of penicillin and streptomycin (Invitrogen) for HL-60 and MV4-11, and Iscove’s modified Dulbecco medium (IMDM) containing 20% fetal bovine serum and 100 units/mL of penicillin and streptomycin (Thermofisher, Waltham, MA, USA) for KG1. Cells were knocked down for LXRβ or VPS34 and control cells were generated and cultured as previously described [1]. Frozen samples from AML patients were obtained after informed consent and stored at the HIMIP collection (BB18 0033-00060). Control peripheral blood mononuclear cells (PBMC) were obtained from three healthy volunteers, age- and sex-matched with AML patients. According to the French law, HIMIP collections were declared to the Ministry of Higher Education and Research (DC 2008-307 collection 1) and obtained a transfer agreement (AC 2008-129) after approbation by the “Comité de Protection des Personnes Sud-Ouest et Outremer II” (ethical committee). Clinical and biological annotations of the samples have been declared to the CNIL (Comité National Informatique et Libertés, i.e., Data processing and Liberties National Committee); samples were obtained from patients diagnosed with AML at the Toulouse University Hospital (TUH) after signed informed consent in agreement with the Declaration of Helsinki. Peripheral blood or bone marrow samples were frozen in fetal calf serum with 10% dimethyl sulfoxide (DMSO) and stored in liquid nitrogen. PBMC and AML primary samples were cultured in Iscove’s modified Dulbecco medium (IMDM) containing 20% fetal bovine serum. For some experiments, fresh leukemic blasts recovered at diagnosis were immediately treated with ethanol, Ara-C 50 µM, DDA 2.5 µM, or both DDA and Ara-C for 48 h. In other cases, frozen cells were thawed in IMDM medium with 20% FBS. Patient characteristics are shown in Appendix A. The in vivo assays were performed with cryopreserved cells.

### 2.3. Lentiviral Infection of KG-1 and HL-60 Cells

Lentiviral particles were generated by calcium phosphate transient transfection in 293T cells. Briefly, 293T into a 10 cm dish were transfected with 62.5 µL CaCl_2_ (2M), 500 µL HeBS 2×, 418 µL H_2_O, 3.5 µg pVSV-G (env), 6.5 µg p8.1 (tat, pol, rev, gag) and 10 µg inducible sh RNA against VPS34 (TRIPZ Human PIK3C3, clone V3THS_372038, GE Healthcare) or Atg 5 and Atg 12 (TRIPZ Human Atg 12, clone V3THS_391721, GE healthcare, Chicago, Il, USA). Then, 72 h after cell transfection, 2 mL of supernatants containing virus were collected and were added to KG-1 or HL-60 cells in a 6-well plate. Polybrene was added at 8 µg/mL final concentration and spinoculation was performed by centrifuging cells 45 min at 800 × *g*. Then, 72 h after transduction, a medium-containing virus was removed and changed for a virus-free medium. After an additional 24 h, cells were selected with 1 µg/mL puromycin. When puromycin-resistant cells appeared, KG-1 or HL-60 expressing high levels of Sh RNA (RFP positive cells) were sorted by flow cytometry after 24 h treatment with 1 µg/mL doxycycline. All Sh RNA experiments were performed on the cell bulk, treated, or not, for 72 h with 1 µg/mL doxycycline for Sh RNA induction. The expression of ATG5, ATG12 and VPS34 was controlled by Western blot (Appendix A).

### 2.4. Cell Death Assay

Cell death was determined using trypan blue assay as previously described. Briefly, 500,000 cells were treated with solvent vehicle (0.1% ethanol), 0.1 µM Ara-C, 5 µM DDA alone or in combination with Ara-C for 48 h. Cells were mixed and 10 µL of resuspended cells was added to 10 µL of a trypan blue solution (0.25% (*w/v*) in phosphate buffer saline (PBS)) and counted in a Malassez cell under light microscopy. Cell viability was then quantified by counting viable (trypan blue-negative) and dead cells (trypan blue-positive).

### 2.5. Annexin-V/7AAD Assay

Apoptosis was determined by flow cytometry (Cytoflex Flow Cytometer; Beckman Coulter, Brea, CA, USA) following double staining, with either annexin V fluorescein isothiocyanate (BD Pharmagen, Franklin Lakes NJ, USA, clone 2331) or 7-aminoactinomycin D (7-AAD). Briefly, 500,000 cells were either treated with ethanol, 0.1 µM Ara-C, 5 µM DDA, alone or in combination for 48 h, then washed with cold phosphate-buffered saline (PBS), and resuspended in 200  µL of Annexin-V binding buffer. Annexin-V-fluorescein isothiocyanate and 7AAD were then added, and the samples were incubated in the dark at room temperature for 15 min. The percentage of viable cells, Annexin-V-7-AAD-negative cells, was scored using a flow cytometer.

### 2.6. Western Blot

Proteins were separated using 4% to 12% polyacrylamide SDS-PAGE gels (Life Technologies, Carlsbad, CA, USA) and electro-transferred onto a 0.2 µM nitrocellulose membrane (GE Healthcare, Chicago, IL, USA). After blocking in Tris-buffered saline (TBS) with 0.1% Tween and 5% bovine serum albumin, membranes were blotted overnight at 4 °C with appropriate primary antibodies. Primary antibodies were detected using appropriate horseradish peroxidase-conjugated secondary antibodies. Immunoreactive bands were visualized by enhanced chemiluminescence (Clarity ECL BIORAD, Hercules, CA, USA) with a Syngene camera. Densitometric analyses of immunoblots were performed using the GeneTools software (Syngene, Cambridge, UK).

### 2.7. Autophagy

The Cyto-ID autophagy dye was used for the quantification of autophagy following instructions provided in the manufacturer’s manual. Briefly, 500,000 cells were collected after treatment with the solvent vehicle (0.01% ethanol), 0.1 µM Ara-C and 5 µM DDA alone, or in combination for 16 h. PBS with 1% FBS was allowed to warm at 37 °C. The working concentration of Cyto-ID Green autophagy dye solution was prepared by mixing 10 µL of the dye and 1 mL PBS with 1% FBS. Cells were washed twice and adjusted to a final concentration of 500,000 cells/mL. After centrifugation for 5 min at 1200 rpm, 100 µL of the working dye was added to each cell pellet, resuspended, and incubated for 30 min at 37 °C, followed by a wash and resuspension with 200 µL of PBS 1× before flow cytometry analysis.

### 2.8. Sterol Analysis

Sterols in cell homogenates were extracted with a solvent mixture containing chloroform/methanol 2/1 (*v/v*), spiked with epicoprostanol as the internal standard. Lipids were partitioned in chloroform after the addition of saline and saponified by methanolic potassium hydroxide (0.5 N, 60 °C, 15 min). The fatty acids released were methylated with BF3-methanol (12%, 60 °C, 15 min), in order to not interfere with the chromatography of sterols. The sterols were re-extracted in hexane and silylated, as described previously [1]. The trimethylsilylether derivatives of sterols were separated by gas chromatography (GC) (Hewlett–Packard 6890 series) in a medium polarity capillary column RTX-65, (65% diphenyl 35% dimethyl polysiloxane, length 30 m, diameter 0.32 mm, film thickness 0.25 μm) (Restesk, Evry, France). The mass spectrometer (Agilent 5975 inert XL, Agilent, Santa Clara, CA, USA) in series with the GC was set up for the detection of positive ions. Ions were produced in the electron impact mode at 70 eV. Sterols were identified by the fragmentogram in the scanning mode and quantified by selective monitoring of the specific ions after normalization with the internal standard epicoprostanol and in calibration with weighed standards.

### 2.9. Phosphatidyl Ethanolamine (PE) Analysis

PE from total cell membranes were extracted, analyzed and quantified by Liquid chromatography-mass spectrometry/mass spectrometry (LC-MS/MS), essentially as described in Lamaziere et al. [28]. Briefly, after the addition of an internal standard of PE, total lipids were extracted from cells according to Bligh and Dyer [29]. Lipid extracts were suspended in the mobile phase buffer containing Isopropanol/Hexane/Ammonium acetate 15 mM (58/40/2). Prior to MS, lipid species were separated using a vinyl alcohol polymer coating HPLC silica column (YMC-Pack PVA-Sil). The quantitation of PE was performed by Electrospray ionization-mass spectrometry/mass spectrometry (ESI-MS/MS) (Qtrap 6500T.M., A.B. Sciex, Les Ulis, France), using a multiple reaction monitoring mode (MRM) and applying correction factors given by internal/external standard calibrations. Full details are described by Wolf and Quinn [30].

### 2.10. Animals

Mice were handled and cared for according to the ethical guidelines of our institution and following the Guide for the Care and Use of Laboratory Animals (National Research Council, 1996) and the European Directive EEC/86/609, under the supervision of authorized investigators (see Supplemental Experimental Procedures). All mice were maintained in specific pathogen-free conditions and were only included in protocols following 2 weeks of quarantine. NSG mice (NOD/LtSz-scid IL2Rγc null), NOD/SCID mice (NOD.CB17-Prkdcscid/), Nude mice (NU/NU) and C57Black6 mice were from Charles River Laboratories, Saint-Germain-sur-L’Arbresle, France.

### 2.11. Measure of Combination Therapy Efficacy In Vivo

Exponentially growing cells were harvested, washed twice in PBS, then resuspended in PBS. HL60, KG1, MV4-11 cell lines (2 × 10^6^ cells) were inoculated subcutaneously into the flanks of NOD/SCID mice. When tumors were palpable, mice were treated. Tumor volume (tumor length × width² × 0.5236) was measured every 2–3 days using calipers. At the end of the experiment, mice were sacrificed, tumors excised, and the volume and mass of the tumors were measured. Human primary AML cells were transplanted into NSG mice, as reported previously [1]. Briefly, mice were housed in sterile conditions using HEPA filtered micro-isolators and fed with irradiated food and sterile water. Transplanted mice were treated with antibiotics (Baytril, Bayer, Germany) for the duration of the experiment. Mice (6–9 weeks old) were sub-lethally treated with busulfan (30 mg/kg) 24 h before injection of leukemic cells. Leukemia samples were thawed at room temperature, washed twice in PBS, and suspended in Hanks balanced salt solution at a final concentration of 1–5 million cells per 100 μL. If no signs of distress were seen, mice were initially analyzed for engraftment 10 weeks after injection, except where otherwise noted. After AML cell transplantation and when mice were engrafted (tested by flow cytometry on peripheral blood), a daily intraperitoneal injection of 10 mg/kg of Ara-C for 5 days and/or 20 mg/kg/day of DDA for 20 days was performed in NSG transplanted recipients. For negative controls, NSG mice were treated daily with an IP injection of vehicle, PBS 1×. Mice were monitored for toxicity and provided nutritional supplements as needed.

### 2.12. Assessment of Leukemic Engraftment

NSG mice were humanely killed in accordance with European ethic protocols. Bone marrow (mixed from tibias and femurs) and spleen were dissected in a sterile environment and flushed in Hanks balanced salt solution with 1% FBS. MNCs from bone marrow and spleen were labeled with FITC conjugated anti-hCD3, PE-conjugated anti-hCD33, PerCP-Cy5.5-conjugated anti-mCD45.1, APC-conjugated anti-hCD45 and PeCy7-conjugated anti-hCD44 (all antibodies from BD (Becton Dickinson, Franklin Lakes, NJ, USA), except FITC-conjugated anti-hCD3 from Ozyme Biolegend ‘Ozyme, Saint-Cyr l’ecole, France)), to determine the fraction of human blasts (hCD45+hCD33+mCD45.1-cells) using flow cytometry. After 3 weeks of treatment, mice were sacrificed and viable murine haematopoietic cells were measured by flow cytometry as Annexin V- CD45− CD45.1+. Analyses were performed on a Beckman coulter cytoflex flow cytometer. The number of AML cells/µL of peripheral blood and number of AML cells in total cell tumor burden (in bone marrow and spleen) were determined using CountBright beads (Invitrogen, Waltham, MA, USA) with the described protocol.

### 2.13. Statistical Analysis

Significant differences in the quantitative data between the control and treated groups were analyzed using a non-parametric one-way or two-way Mann–Whitney test (GraphPad Prism, version 7.0, Graphpad Software, San Diego, CA, USA). *p* values of less than 0.05 were considered to be significant (* *p* < 0.05, ** *p* < 0.01 and *** *p* < 0.001).

## 3. Results

### 3.1. DDA Potentiates Ara-C Cytotoxicity in AML Cell Lines

The cytotoxic activity of DDA and Ara-C alone or in combination was studied on three leukemia cell lines (HL-60, MV4-11 and KG1). DDA activity, in combination with Ara-C, was assessed using drug concentrations extrapolated from individual IC_50_ values. The combined treatment of DDA with Ara-C shows a 20% increase in cell death in co-treated conditions compared to cells treated with DDA and 50% compared to cells treated with Ara-C (Figure 1A–C). The combinatorial effect on cytotoxicity was assessed by the calculation of a combinatorial index (CI) value across a range of drug concentrations, using the Chou-Talalay method. The calculated combination index (< 1) shows that DDA synergized with Ara-C to kill HL-60 (Figure 1A), MV4-11 (Figure 1C), and KG1 cells (Figure 1E). As an illustration, we report in Figure 1B,D,F that co-treatment using 5 µM DDA and 0.1 µM Ara-C for 48 h potentiated cytotoxicity in the three tested cell lines.

### 3.2. Single and Combination DDA/Ara-C Treatments Induce Characteristics of Autophagy in AML Cell Lines

We next analyzed AML cell lines treated with 5 µM DDA and 0.1 µM Ara-C, alone or in combination, for the presence of autophagy characteristics. Single and combination treatments increased the formation of acidic vesicles labeled by the Cyto-ID fluorophore in HL-60 and KG1 cells, while no significant labeling was observed in solvent-vehicle treated control cells (Figure 2A). This suggests that drugs alone, and in combination, induce the formation of autophagosomes/autolysosomes. This increase was associated with LC3-II, the lipidated form of LC3 (Figure 2B), and autophagosomes’ formation (Figure 2C). We previously showed that the accumulation of Δ8-sterol (zymostenol and 8-dehydrocholesterol) due to the inhibition of the 3β-hydroxysteroid-Δ8,Δ7-isomerase (EBP, D8D7I) by DDA cooperated with the LXR-dependent expression of pro-autophagic genes by DDA to induce lethal autophagy [1,12,14]. We thus determined the sterol profile of cells treated with drugs alone or in combination. We showed that DDA alone or in combination with Ara-C equally induced the accumulation of delta-8-sterols, while Ara-C alone has no impact on the post-lanosterol cholesterol biosynthetic pathway (Figure 2D). LXR was shown to control the expression of the CTP: Phosphoethanolamine Cytidyltransferase enzyme (Pcyt2), which produces phosphatidylethanolamine (PE) [31], the lipid involved in LC3-I lipidation, leading to the active LC3-II [32,33]. Taking into account the impact of DDA and Ara-C on the lipid metabolism [1,34,35,36], we next investigated the impact of DDA alone or in combination with Ara-C on intracellular phosphatidylethanolamine (PE) levels. We found that single or combination treatments increased PE levels in AML cells. (Figure 2E). We observed a potentiation in the production of PE for combination treatments in both cell lines. Together, these data showed that combination treatments using DDA with Ara-C potentiate the appearance of key autophagic events in AML cells.

### 3.3. Autophagy Is Responsible for the Potentiation of Ara-C Cytotoxicity by DDA in AML Cell Lines

DDA was shown to kill cancer cells by lethal autophagy [1] and we showed, in the above section, the synergistic cytotoxic effects resulting from the combination of 5 µM DDA and 0.1 µM Ara-C. We then studied whether the potentiation of cell death observed during combination treatments was driven by autophagy or by apoptosis. While the pharmacological inhibition of caspase-dependent cell death decreased Ara-C cytotoxicity in HL-60 (*p* < 0.01, *n* = 5) and KG1 cells (*p* < 0.001, *n* = 5), it did not inhibit the cytotoxicity induced by DDA alone or in combination with Ara-C in HL-60 and KG1 cell lines (Figure 3A). By contrast, the pharmacological (Baf A1) and genetic (shATG5, shATG12, shVPS34) inhibitions of autophagy decreased cytotoxicity in single and combination treatments (Figure 3B,C and Appendix A), while it increased Ara-C cytotoxicity in HL-60 cells (Figure 3B,C).

### 3.4. DDA Drives the Potentiation of Ara-C Cytotoxicity in AML Cell Lines in A LXRβ-Dependent Manner

We previously showed that the nuclear receptor LXRβ was required for DDA-induced lethal autophagy [1]. We show here that the LXR agonist GW3965 inhibits DDA cytotoxicity alone or in combination with Ara-C, while it does not inhibit the cytotoxicity induced by a single Ara-C treatment (Figure 4A and Appendix A). The importance of LXRβ in DDA and DDA/Ara-C treatments was further confirmed using KG1 cells, in which LXRβ expression was invalidated using a single hairpin RNA approach (Figure 4B). We then found that the increase in intracellular PE levels induced by DDA alone and in combination with Ara-C was inhibited by the LXRβ agonist GW 3965, while GW3965 had no impact on the increase of PE levels induced by Ara-C alone. The increase in PE level by DDA was not observed in KG1 cells knocked down for LXRβ (Figure 4C,D). Taken together, these data showed that lethal autophagy drives the potentiation of Ara-C cytotoxicity by DDA through LXRβ, and that the stimulation of PE levels by DDA was LXRβ-dependent in single DDA and DDA/Ara-C treatments.

### 3.5. DDA Potentiates Ara-C Cytotoxicity in Primary AML from Patients

We then tested the sensitivity of a panel of 20 primary tumors from AML patients with different cytogenetic and molecular characteristics (Appendix A), to DDA and Ara-C, alone or in combination. We observed that combination treatments significantly reduced cell viability in the leukemic bulk (Figure 5A). Indeed, the leukemic bulk viability was reduced by 38% in DDA-Ara-C treated cells compared to cells treated with DDA, by 15% compared to Ara-C treated cells, and by 88% compared to vehicle (Figure 5A). Interestingly, DDA/Ara-C combination treatments sensitize AML cells to Ara-C in the Ara-C-resistant subgroups (Figure 5B). In addition, it seems that no difference in the response to treatment was found in relation to cytogenetic risk or FLT3/NPM1 mutations, even though too few patients were analyzed (Figure 5C–E). We also tested the toxicity of the co-treatment on normal peripheral blood mononuclear cells (PBMC), and on normal lymphocytes showing (Figure 5F) that the toxicity of combination treatments was not significantly different from that of single treatments. In addition, combination treatments were less cytotoxic for normal PBMC and lymphocytes (Figure 5F) compared to primary AML samples (Figure 5A). These results show that DDA sensitizes primary AML, but not normal samples to Ara-C. We then sought to determine whether combination treatments induced the characteristics of autophagy in AML patient samples. The treatment of primary AML cells with DDA induced the appearance of macro-vesicles as previously described [1]. Ara-C displayed only a weak effect on the appearance of vesicles, while DDA/Ara-C co-treatment potentiated the intracellular accumulation of macro-autophagic vesicles (Figure 5G), consistent with a high expression of LC3-II (Figure 5H).

### 3.6. DDA Potentiates the Anti-Leukemia Effects of Ara-C In Vivo in AML Cell Lines and Patient-Derived AML Xenografted in Mice

We next evaluated whether DDA could sensitize and potentiate AML cell lines to Ara-C in vivo. HL-60, KG1 and MV4-11 cells were implanted into NOD/SCID mice and treated with compounds, as described in Figure 6A. DDA treatment induced a significant control of tumor growth (20 mg/kg/day), in KG-1, HL-60 and MV4-11 tumors (Figure 6B–D). The daily administration of Ara-C (10 mg/kg/days) had no significant impact on tumor growth (volume and weight) (Figure 6B–D), while co-treatment of DDA/Ara-C potentiated the growth control of tumors in the three xenografted cell lines (Figure 6B–D). These data established that the potentiation of the cytotoxic effect of Ara-C by DDA observed in vitro was also measured in vivo. To evaluate the relevance of these observations in patient tumors, we next investigated the efficacy of combination treatments on primary AML cells from three patients that were orthotopically engrafted in NSG (NOD/LtSz-scid IL2Rγc null) mice. After engraftment, mice were treated with DDA (20 mg/kg/day by intraperitoneal (i.p.) injection for 20 days) and Ara-C (10 mg/kg/day by i.p. injection for 5 days) (Figure 6E). In these conditions, combination treatments drastically improved the efficacy of Ara-C, with a mean efficacy of 90% for the three tested patient-derived tumors (Figure 6F). These data revealed that combination treatments of DDA with Ara-C sensitize AML cell lines and primary AML from patients in vivo, compared to Ara-C alone. We measured a significant expansion of the murine compartment after combination treatment (Figure 6G), showing that treatment was well tolerated.

## 4. Conclusions

In this study, we report that the newly identified tumor suppressor and cholesterol metabolite DDA potentiated and sensitized in vitro and in vivo leukemic cells lines and primary AML samples to a conventional chemotherapeutic agent Ara-C (cytarabine), which is widely used in the clinic for AML treatment. Importantly, combination tests performed on 20 genetically diverse primary AML samples revealed a similar efficacy of DDA/Ara-C combination treatments. Furthermore, our data showed that the potentiation and sensitization due to the DDA/Ara-C combination treatment was driven by the lethal autophagy triggered by DDA in an LXRβ-dependent manner.

LXRβ is a ligand-dependent transcription factor that is strongly expressed in AML cells [1]. We previously showed that DDA killed AML cells via LXRβ, and this effect was not observed with canonical LXR ligands, such as 22(R)-hydroxycholesterol, TO0901317 or GW3965, highlighting an originality in DDA action compared to other LXR ligands [1,14]. DDA behaves like a biased agonist of LXRβ. DDA does not activate canonical LXR-dependent genes expression such as ABCA1, but specifically stimulates the expression of lysosome biogenesis- and autophagy-encoding genes through the activation of LXRβ [1,12,13,14]. In addition, DDA, but not canonical LXR ligands, induces the accumulation of pro-autophagic Δ8 sterols, due to the inhibition by DDA of ChEH, and in particular, its D8D7I subunit. LXR has been proposed as an anticancer target, as LXR controls cholesterol homeostasis [37,38], which is known to be deregulated in most cancers, including AML [39,40,41]. Interestingly, LXR agonists TO0901317 or GW3965, in combination with bexarotene, a retinoid X receptor ligand, were reported to induce differentiation and apoptosis in AML cells [42]. Here, we show that LXRβ can be driven by another class of LXR ligand to trigger lethal autophagy in AML cells, and sensitizes them to Ara-C treatment in vitro and in vivo.

We report here, for the first time, that DDA and Ara-C individually increased PE levels in AML cells and that combination treatments synergized in increasing the PE production in these cells. PE is the lipid involved in LC3-I lipidation leading to functional LC3-II, which is required for autophagosome formation [32,33]. PE accumulation is thus a new important parameter to be considered in autophagy processes induced by DDA and Ara-C [1,23,43]. We found here, to our surprise, that low doses (0.1 µM) of Ara-C significantly increased the intracellular levels of PE. This increase is not mediated by LXRβ and the mechanism involved in this effect deserves further investigations. We showed that DDA increased PE levels in AML cells and that this effect was LXRβ-dependent. Importantly, the potentiation of the cytotoxicity of DDA/Ara-C co-treatments also potentiates the increase in PE, which may sustain the autophagic flux through the maintenance of LC3 lipidation. The increase in PE levels was proposed as a key event required to activate autophagic processes [44,45], and this hypothesis is supported for the first time by the present study.

One of the causes of acquired resistance to chemotherapy is the activation of pro-survival autophagy by drugs. This led to the development of strategies aiming to inhibit autophagy in order to improve the therapeutic outcome in patients [24,25,26]. Here, we show that the combination of a compound that induces lethal autophagy to a chemotherapeutic drug, that induces protective autophagy, can synergize to kill AML cells by lethal autophagy. The potentiation and sensitization of AML to Ara-C treatment represents a promising therapeutic option that deserves further investigation in clinical trials.

## Figures and Tables

**Figure 1 cancers-12-01725-f001:**
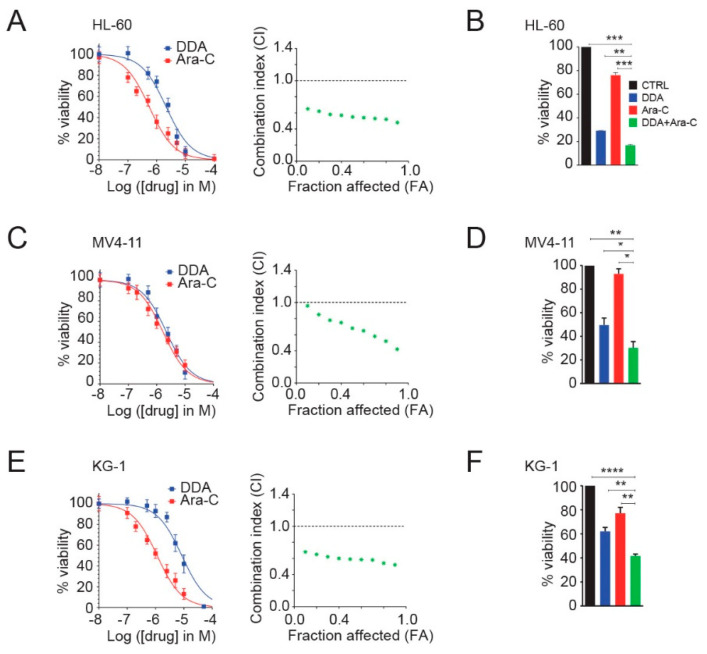
Dendrogenin A (DDA) synergizes with antimetabolite cytarabine (Ara-C) to reduce proliferation and to kill AML cells. HL-60 cells (**A**) were treated with DDA (0–100 µM) and Ara-C (0–10 µM) for 48 h. Cell viability was measured by the Trypan Blue exclusion method and reported on the left. Bars are mean ± SEM of 5 independent experiments. On the right graph, CI values resulting from different combination tests performed with various concentrations of DDA and Ara-C were calculated according to the Chou-Talalay method. The dashed line designates a CI value of 1, with CI < 1 being synergistic, CI = 1 being additive, and CI >1 being antagonistic. Data are representative of three independent experiments. (**B**) Cell viability of HL-60 cells treated for 48 h with 5 µM DDA; 0.1 µM Ara-C alone or in combination was measured by the Trypan Blue exclusion method. Bars are mean ± SEM of five independent experiments. Similar experiments were conducted with MV4-11 (**C**,**D**) and KG1 cells (**E**,**F**). Uncropped Western Blot Figures could see Appendix A. * *p* < 0.05, ** *p* < 0.01, *** *p* < 0.001, **** *p* < 0.0001, n.s: non significant.

**Figure 2 cancers-12-01725-f002:**
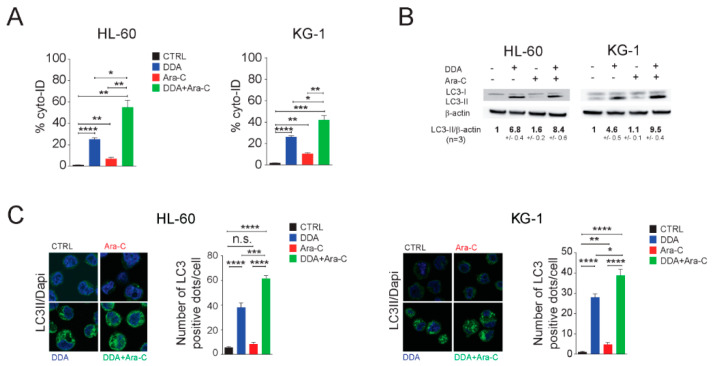
DDA/Ara-C Combination treatment induced characteristics of autophagy. HL-60 and KG1 cells were treated for 16 h with 5 µM DDA or 0.1 µM Ara-C, alone or in combination. (**A**) The presence of acidic vesicles was revealed using Cyto-ID. Data were represented as the % of Cyto-ID-positive cells, in Annexin-V-/7AAD- cells, relative to control cells. (**B**) The expression of LC3 proteins in treated cells was analyzed by western blot. Numbers represent the LC3II/actin ratios obtained by densitometric analysis (*n* = 5 ± SEM). Uncropped Western Blot Figures could see Appendix A. (**C**) LC3 dots were measured by immunocytochemistry and quantification of LC3 dots are reported (*n* = 5 ± SEM). (**D**) Results from the quantification of steroidal cholesterol precursors in treated cells is reported: lano: lanosterol, zyn: zymostenol; zyr: zymosterol; lath: lathosterol; 8-D: 8-dehydrocholesterol; 7-D: 7-dehydrocholesterol; des: desmosterol; chol: cholesterol. (**E**) Results from the quantification of phosphatidylethanolamine (PE) in treated cells is reported. Data are the mean ± SEM of five independent experiments performed in triplicate. * *p* < 0.05, ** *p* < 0.01, *** *p* < 0.001, **** *p* < 0.0001, n.s: non significant.

**Figure 3 cancers-12-01725-f003:**
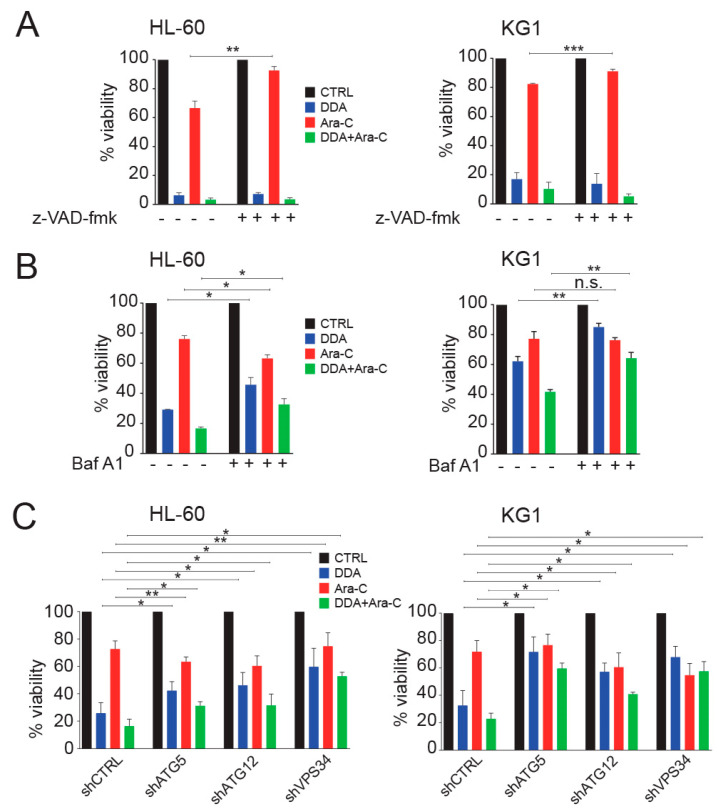
Autophagy drives the potentiation of Ara-C cytotoxicity by DDA. HL-60 and KG1 cells were treated for 48 h with 5 µM DDA and 0.1 µM Ara-C, alone or in combination, in the absence or presence of (**A**) 40 µM z-VAD-fmk or (**B**) 5 nM bafilomycin A1 (Baf A1). Viability was measured by the Trypan Blue exclusion method. Bars are mean ±SEM of five independent experiments. (**C**) HL-60, and KG1 cells transfected with control shRNA (shCTRL) or shRNA against ATG5 (shATG5), ATG12 (shATG12) and against VPS34 (shVPS34) were treated for 48 h with 5 µM DDA and 0.1 µM Ara-C, alone or in combination. Viability was measured as described above. Bars are mean ± SEM of five independent experiments. * *p* < 0.05, ** *p* < 0.01, *** *p* < 0.001, n.s: non significant.

**Figure 4 cancers-12-01725-f004:**
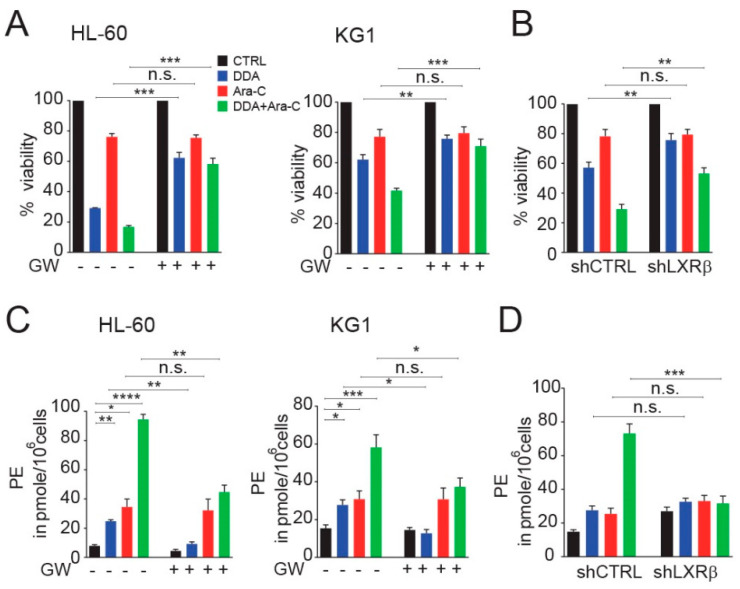
LXRβ controls the potentiation of Ara-C cytotoxicity and PE production by DDA. (**A**) HL-60 and KG1 cells were treated for 48 h with 5 µM DDA and 0.1 µM Ara-C, alone or in combination, in the absence or presence of 2 µM GW3965. Viability was measured by the Trypan Blue exclusion method. (**B**) KG1 cells transfected with control shRNA (shCTRL) or shRNA against LXRβ were treated for 48 h with 5 µM DDA and 0.1 µM Ara-C, alone or in combination. Viability was measured as described above. (**C**) HL-60 and KG1 cells were treated for 48 h with 5 µM DDA and 0.1 µM Ara-C, alone or in combination, in the absence or presence of 2 µM GW3965. Results from the quantification of phosphatidylethanolamine (PE) in treated cells are reported. (**D**) KG1 cells transfected with control shRNA (shCTRL) or shRNA against LXRβ were treated for 48 h, with 5 µM DDA and 0.1 µM Ara-C, alone or in combination. Results from the quantification of phosphatidylethanolamine (PE) in treated cells are reported. Bars are mean ± SEM of five independent experiments. * *p* < 0.05, ** *p* < 0.01, *** *p* < 0.001, **** *p* < 0.0001, n.s: non significant.

**Figure 5 cancers-12-01725-f005:**
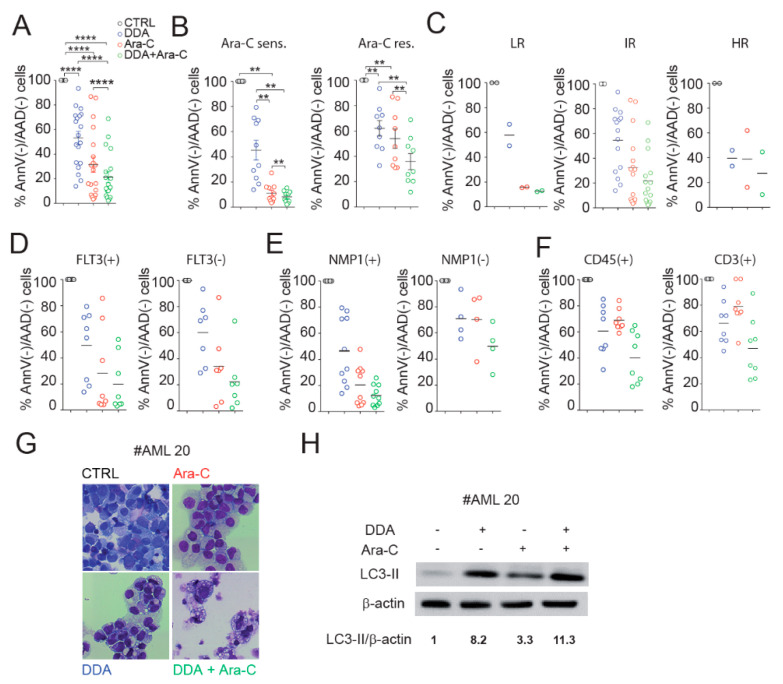
DDA potentiates Ara-C cytotoxicity in primary acute myeloid leukemia (AML) cells from patients. Samples from AML patients (*n* = 20) were treated for 48h with 2.5 µM DDA or 50µM Ara-C alone or in combination. (**A**) Cell death was assessed in the leukemic bulk (CD45+) using AnnexinV/7AAD staining. Data were represented as % of survival corresponding to Annexin-V-/7AAD- cells relative to total cells. Different representations of the results are shown taken into account the sensitivity to Ara-C (**B**); Ara-C sens.: Ara-C sensitive (mean < 50% Annexin-V-/7AAD- cells at 50µM Ara-C); Ara-C res.: Ara-C resistant (mean > 50% Annexin-V-/7AAD- cells at 50 µM Ara-C); the prognostic risk category(**C**); LR: low risk, IR: intermediate risk, HR: high risk; the FLT3 (**D**) and the NMP1 (**E**) status. (**F**) PBMC from healthy control samples (*n* = 8) were treated with DDA (2.5 µM) or Ara-C (50 µM) or both DDA and Ara-C (2.5/50 µM), or vehicle for 48h. Cell death was assessed in the leukemic bulk (CD45+) using AnnexinV/7AAD staining. Data were represented as the % of survival corresponding to Annexin-V-/7AAD- cells relative to total cells. (**G**) Representative images of primary AML cells stained with May Grumwald Giemsa after 48 h of treatment with solvent vehicle, 2.5 µM DDA and 50 µM Ara-C, alone and in combination. (**H**) Western blot analysis of the expression of LC3-I and LC3-II in primary AML cells after 24 h of treatment with the solvent vehicle, 2.5 µM DDA and 50 µM Ara-C, alone or in combination. Numbers represent the LC3II/actin ratios obtained by densitometric analysis. Uncropped Western Blot Figures could see Appendix A. ** *p* < 0.01, **** *p* < 0.0001.

**Figure 6 cancers-12-01725-f006:**
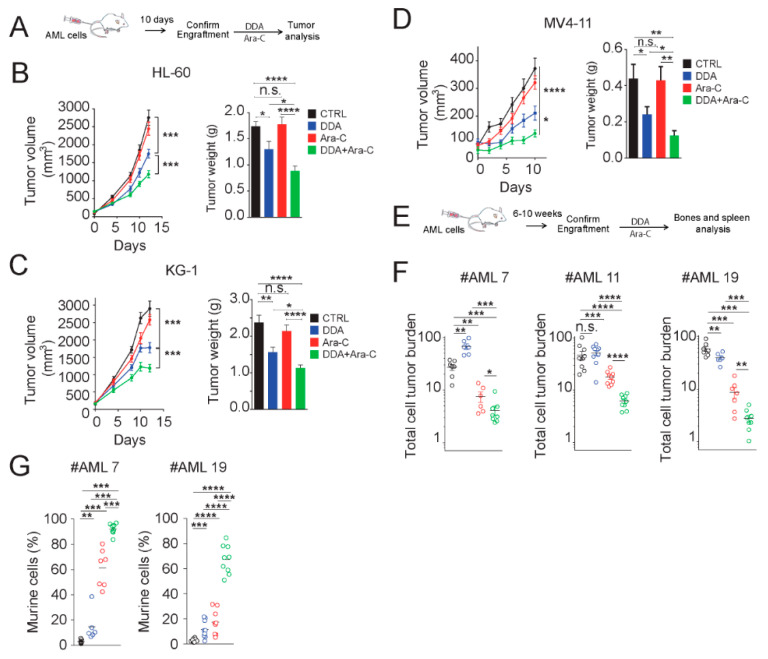
DDA potentiates Ara-C cytotoxicity in AML cell lines and AML from patients in vivo. (**A**) Experimental scheme used in Figure 6B–D. Results of five independent experiments. Tumor volume curve and average tumor weight measured at the end of the experiments of (**B**) HL-60, (**C**) KG1 and (**D**) MV4-11 xenografts (*n* = 10 per group) in NOD/SCID mice treated with DDA (20mg/kg/day by i.p. injection) or Ara-C (10 mg/kg/day for five days by i.p injection) or both DDA and Ara-C or vehicle control. (**E**) Experimental scheme used in Figure 6F, wherein primary AML cells from patients were injected intravenously into NSG mice (three different AML patients were tested). After the validation of tumor engraftment, mice were treated with DDA (20 mg/kg/day by i.p. injection for 20 days) or Ara-C (10 mg/kg/day for five days by i.p. injection) or both DDA and Ara-C or vehicle control (**F**). Human leukemic cell content in bone marrow and spleen was measured by flow cytometry using human anti-CD45 and human anti-CD33 antibodies. (**G**) Measure of murine cell contingent expansion. * *p* < 0.05, ** *p* < 0.01, *** *p* < 0.001, **** *p* < 0.0001, n.s: non significant.

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
