# Peer review of "Dendrogenin A Synergizes with Cytarabine to Kill Acute Myeloid Leukemia Cells In Vitro and In Vivo"

_cancers, 2020, doi:10.3390/cancers12071725_

Round 1

Reviewer 1 Report

The study of Shernan et al has investigated the combined effects of DDA which affects the cholesterol metabolism and cytarabine by using in vitro and in vivo assays. They demonstrate that the combined use is more effective in killing some AML cell lines. Autophagy might be one mechanism that is affected. Although the study is of interest, a number of items can be approved.

-It will be of interest to show whether the observed effects of DDA and Ara-C is selective for the malignant counterpart or that normal CD34+ cells are also affected by using in vitro colony assays. Alternatively, a difference in dosing might exist between normal vs AML CD34+ cells. These items are relevant for the clinical application of the combined use of Ara-C and DDA. The use of PBMNC and survival assays are of limited value in this context. Similar applies to lymphocytes.

-In fig 1 the effects of DDA, Ara-C and combined use is shown but a great variability is observed. In fig 1B only 20% of HL60 cells survive in the presence of DDA which makes it impossible to study the combined use. In fig 1 D and F almost no effect is shown of Ara-C. It is unclear why the authors use this dose and whether the variability in effects correlates with the number of cells in S-phase?

-For studying autophagy it is in general recommended to use 3 different assays (See Guidelines for the use and interpretation of assays for monitoring autophagy (3rd edition). It would be attractive for the cell lines to use the mCherry-GFP-LC3 construct.

-In fig 2 the results of PE accumulation are shown. The interpretation of the findings are difficult to understand since Ara-C and DDA induced comparable results which makes it unlikely that it correlates with autophagy. The authors used for this assay viable cells after 16 hrs. of exposure? The setup of the experiment might be relevant since almost no cells survive after 48 hrs. of the combined use of Ara-C and DDA.

-In fig 3 Z-VAD-fmk is used showing that this compound has no effect on the survival of the cells. The authors have to prove that they used an effective dose of this compound to block caspase activation.

In addition the results of knock-out studies for shATG5, shATG12, and shVPS34 are shown. A number of studies have shown that ATG genes are critical for the survival of normal CD34+ and leukemic cells. It is unclear from the experiments whether the survival of the control cells were already strongly affected by k.o. of the ATG gene which has been shown by previous publications. In addition the authors have to demonstrate the degree of downregulation as induced by sh approach by using PCR or western blotting.

-To study the leukemic counterpart and especially the leukemic stem cell enriched fraction it is questionable whether survival assays after 48 hrs of exposure have any relevance. Therefore follow-up experiments should be performed with AML CD34+ cells by using in vitro colony assays or long-term culture assays.

-The results of the in vivo experiments are of interest. Can the authors present the results with Kaplan Meier curves to demonstrate that the mice have a survival advantage in the presence of the different compounds.

Author Response

The study of Shernan et al has investigated the combined effects of DDA which affects the cholesterol metabolism and cytarabine by using in vitro and in vivo assays. They demonstrate that the combined use is more effective in killing some AML cell lines. Autophagy might be one mechanism that is affected. Although the study is of interest, a number of items can be approved.

-It will be of interest to show whether the observed effects of DDA and Ara-C is selective for the malignant counterpart or that normal CD34+ cells are also affected by using in vitro colony assays. Alternatively, a difference in dosing might exist between normal vs AML CD34+ cells. These items are relevant for the clinical application of the combined use of Ara-C and DDA. The use of PBMNC and survival assays are of limited value in this context. Similar applies to lymphocytes.

Authors response:

Thank you for this comment. DDA alone have been shown to display an anti-leukemic effect on CD34+CD38- immature blasts (Segala et al, Nat commun, 2017). We have addressed this question indirectly showing that co-treatment DDA + Ara-C on primary tumors implanted on NGS mice is associated with the expansion of normal murine CD34+ cells showing the weak toxicity of the combination treatment on normal cells.

We have added the figure 6G (see attached document)

The text was modified as follows: " We measured a significant expansion of the murine compartment after combination treatment (Fig 6G) showing that treatment was well tolerated".

The legend was modified as follows: ". (G) Measure of murine cell contingent expansion in treated mice."

-In fig 1 the effects of DDA, Ara-C and combined use is shown but a great variability is observed. In fig 1B only 20% of HL60 cells survive in the presence of DDA which makes it impossible to study the combined use. In fig 1 D and F almost no effect is shown of Ara-C. It is unclear why the authors use this dose and whether the variability in effects correlates with the number of cells in S-phase?

Authors response:

the combination treatment using 5 µM of DDA and 0.1µM Ara-C is significantly different and not too strong to enable the measurement of different biochemical parameters to be done.

-For studying autophagy it is in general recommended to use 3 different assays (See Guidelines for the use and interpretation of assays for monitoring autophagy (3rd edition). It would be attractive for the cell lines to use the mCherry-GFP-LC3 construct.

Authors response:

We have added IHC analyses of endogenous LC3 punctate formation using an anti-LC3 antibody. Results are presented on figure 2.(see attached document)

The legend has been modified as follows:

"(C) LC3 dots were measured by immunocytovhemistry and quantification of of LC3 dits are reported (n=5 +/- SEM) "

"The text was modified as follows:

" the lipidated form of LC3 (Fig 2B), and autophagosomes formation (Fig 2C) .

-In fig 2 the results of PE accumulation are shown. The interpretation of the findings are difficult to understand since Ara-C and DDA induced comparable results which makes it unlikely that it correlates with autophagy. The authors used for this assay viable cells after 16 hrs. of exposure? The setup of the experiment might be relevant since almost no cells survive after 48 hrs. of the combined use of Ara-C and DDA.

Authors response:

PE being required for LC3-I lipidation and production of LC3-II. PE dosage were performed on viable cells after 16 h exposure to drugs. At 16 hr exposure to drugs, more than 50% cells are still viable which supports the significance of our data.

-In fig 3 Z-VAD-fmk is used showing that this compound has no effect on the survival of the cells. The authors have to prove that they used an effective dose of this compound to block caspase activation.

Authors response:

We have used the same concentration in z-VAD-FMK that was proved to inhibit PARP proteolysis and caspase 3 maturation in previous experiments (Segala et al, Nature communications, 2017, supplementary figure ..), and we observed here that the caspase inhibitor protects cells against cytarabine toxicity (figure x)

In addition the results of knock-out studies for shATG5, shATG12, and shVPS34 are shown. A number of studies have shown that ATG genes are critical for the survival of normal CD34+ and leukemic cells. It is unclear from the experiments whether the survival of the control cells were already strongly affected by k.o. of the ATG gene which has been shown by previous publications. In addition the authors have to demonstrate the degree of downregulation as induced by sh approach by using PCR or western blotting.

Authors response:

We have not measured more that 10% toxicity on shATG5, shATG12 and shVPS34 HL60 and KG1 cells. We have added western blots showing the decrease in ATG5, ATG12 and VPS34 proteins in sh cells compared to mock cells.:

supplementary figure S1: see attached document

We added a section on the material and method describing the production of  knock down cells:

"2.3. Lentiviral infection of KG-1 and HL-60 cells

Lentiviral particles were generated by calcium phosphate transient transfection in 293T cells. Briefly, 293T into a 10cm dish were transfected with 62,5µl CaCl2 (2M), 500 µl HeBS 2X, 418 µl H2O, 3,5µg pVSV-G (env), 6,5µg p8.1 (tat, pol, rev, gag) and 10µg inducible sh RNA against VPS34 (TRIPZ Human PIK3C3, clone V3THS_372038, GE Healthcare) or Atg 5 and Atg 12 (TRIPZ Human Atg 12, clone V3THS_391721, GE healthcare)12. Then, 72h after cell transfection, 2ml of supernatants containing virus were collected and were added to KG-1 or HL-60 cells in a 6 wells plate. Polybrene was added at 8µg/ml final concentration and spinoculation was performed by centrifuging cells 45min at 800g. 72h after transduction, medium containing virus was removed and changed for a virus-free medium. After additional 24h hours, cells were selected with 1µg/ml puromycin. When puromycin-resistant cells appeared, KG-1 or HL-60 expressing high level of Sh RNA (RFP positive cells) were sorted by flow cytometry after 24h treatment with 1µg/ml doxycycline. All Sh RNA experiments were performed on the cell bulk, treated or not 72h with 1µg/ml doxycycline for Sh RNA induction. The expression of ATG5, ATG12 and VPS34 was controlled by western blot (Figure S1)"

-To study the leukemic counterpart and especially the leukemic stem cell enriched fraction it is questionable whether survival assays after 48 hrs of exposure have any relevance. Therefore follow-up experiments should be performed with AML CD34+ cells by using in vitro colony assays or long-term culture assays.

Authors response:

Unfortunately primary tumor cells from patients do not survive more than 4 days in culture which makes impossible to perform such experiments

-The results of the in vivo experiments are of interest. Can the authors present the results with Kaplan Meier curves to demonstrate that the mice have a survival advantage in the presence of the different compounds.

Authors response:

We cannot perform this analysis because all mice groups were sacrificed at the same time in the case of cell lines implanted in mice. For primary tumors the time required for such analysis is too long because mice from all groups survived more than 1 year, and we were obliged to sacrifice them by our ethical committee.

Reviewer 2 Report

General comments:

In this manuscript, the authors show Dendrogenin A can synergize with Cytarabine in AML cell lines and primary patient samples in vitro and vivo. The experimental concept is not brand new, as Dendrogenin A has been shown can trigger lethal autophagy in AML, and Cytarabine is a standard treatment for AML. But the combination of these two drugs leads to a much strong effect in cell lines and patient samples which may have clinical relevance. The manuscript is well written and logically organized.

Specific comments:

 Quantify western blot signal for figure 2 and 5

Figure 5G Label which patient sample was used for the staining.

Author Response

Reviewer 2

General comments:

In this manuscript, the authors show Dendrogenin A can synergize with Cytarabine in AML cell lines and primary patient samples in vitro and vivo. The experimental concept is not brand new, as Dendrogenin A has been shown can trigger lethal autophagy in AML, and Cytarabine is a standard treatment for AML. But the combination of these two drugs leads to a much strong effect in cell lines and patient samples which may have clinical relevance. The manuscript is well written and logically organized.

Specific comments:

 Quantify western blot signal for figure 2 and 5

Authors response: we have provided a densitometric analysis on figure 2 and 5 (see attached document)

Figure 2:

the figure legend was modified as follows:

(B) The expression of LC3 proteins in treated cells was analyzed by western blot. Numbers represent the LC3II/actin ratios obtained by densitometric analysis (n=5 +/- SEM). "

Figure 5:

the following sentence was added to the legend of figure 5:

" Numbers represent the LC3II/actin ratios obtained by densitometric analysis."

Figure 5G Label which patient sample was used for the staining.

Authors response: the patient sample was labeled as "#AML20"

Reviewer 3 Report

The authors have substantial reports about the efficacy of natural onco-suppressor metabolite, dendrogenin A (DDA), to cancers. In this manuscript, they evaluated the synergistic effect of DDA and Ara-C (cytarabine) to AML cells in vitro and in vivo. In particular, they found the mechanism of this synergistic effect is due to the activation of lethal autophagy contributed from DDA. This manuscript is written very well. Some comments were shown below.

  1. The authors demonstrated that simultaneously administration of DDA (5 μM) and cytarabine (0.1 μM) have a synergistic effect; however, it seems that DAA had major effect and cytarabine had only minor effect in most experiments. Finally, the authors concluded that DDA potentiated and sensitized AML cells to cytarabine in vitro and in vivo. What we question is why not say that cytarabine potentiated and sensitized AML cells to DDA instead.
  2. The authors stated that combination treatment were less cytotoxic for normal PBMC and lymphocytes; this statement is not consistent with data shown in figure 5F and 5G. Figure 5F and 5G demonstrated significant decrease of live cells (% AnnV-/AAD-), such as CD45+ and CD3+ cells, respectively.
  3. Discussion regarding to the mechanism of synergistic effect of DDA and cytarabine (or how DDA and cytarabine interact) is suggested.
  4. Some typo and mistyping in this manuscript, such line107, 191, 230, 316, 336, 394, 398; mistakes in these sites should be revised.

Author Response

Reviewer 3:

The authors have substantial reports about the efficacy of natural onco-suppressor metabolite, dendrogenin A (DDA), to cancers. In this manuscript, they evaluated the synergistic effect of DDA and Ara-C (cytarabine) to AML cells in vitro and in vivo. In particular, they found the mechanism of this synergistic effect is due to the activation of lethal autophagy contributed from DDA. This manuscript is written very well. Some comments were shown below.

  1. The authors demonstrated that simultaneously administration of DDA (5 μM) and cytarabine (0.1 μM) have a synergistic effect; however, it seems that DAA had major effect and cytarabine had only minor effect in most experiments. Finally, the authors concluded that DDA potentiated and sensitized AML cells to cytarabine in vitro and in vivo. What we question is why not say that cytarabine potentiated and sensitized AML cells to DDA instead.

Authors response:

We do it in that sense because Ara-C is the drug used in the clinic, and we thought that the formulation was more convenient like that. We have a chosen a concentration in cytarabine with a minimal effect to determine if we could improve its cytotoxicity with active DDA doses.

  1. The authors stated that combination treatment were less cytotoxic for normal PBMC and lymphocytes; this statement is not consistent with data shown in figure 5F and 5G. Figure 5F and 5G demonstrated significant decrease of live cells (% AnnV-/AAD-), such as CD45+ and CD3+ cells, respectively.

Authors response:

We observed a toxicity less important in PBMC (40%) compared to AML (20%).

We have addressed this question indirectly showing that co-treatment DDA + Ara-C on primary tumors implanted on NGS mice is associated with the expansion of normal murine CD34+ cells showing the weak toxicity of the combination treatment on normal cells.

We have added the figure 6G (see attached document)and the legend was modified accordingly

The text was modified as follows: " We measured a significant expansion of the murine compartment after combination treatment (Fig 6G) showing that treatment was well tolerated".

The legend was modified as follows: ". (G) Measure of murine cell contingent expansion in treated mice."

  1. Discussion regarding to the mechanism of synergistic effect of DDA and cytarabine (or how DDA and cytarabine interact) is suggested.

Authors response:

Our unpublished experiments performed on HL60, KG1 and patients primary tumors suggest that DDA does not affect cytarabine metabolism or efflux at the transcriptional level (see figures). We have compared  the expression of genes involved on cytarabine transport and metabolism at the mRNA level by qPCR on HL-60, KG-1 and pimary AML tumors from patient treated or not by DDA (n=4). Genes anayzed are: Equilibrative nucleotide transporter 1 (ENT1) also known as Solute Carrier Family 29 Member 1(SLC29A1), involved in the uptake of Ara-C , deoxycytidine Kinase (dCK) a rate-limiting Ara-C activating enzyme; the Ribonucleotide Reductase Catalytic Subunit M1 (RRM1), which regulates the intracellular pool of dCTP; 5'-Nucleotidase, Cytosolic II (NT5C2) an Ara-C inactivating enzyme; transporters from the ATP Binding Cassette family (ABCC4 and ABCCC5) both involved in Ara-C efflux. DDA does not seem to modulate the expression of these genes.

We thus supposed that DDA and cytarabine are cooperating through different mechanism on the increase in PE levels which probably saturates LC3 lipidation and autophagic flux which led to cell death.

  1. Some typo and mistyping in this manuscript, such line107, 191, 230, 316, 336, 394, 398; mistakes in these sites should be revised.

Authors response: mistakes and typo errors have been corrected.

Line 107 : “described 1 “ was replaced by “ described [1] “

Line 191 : “(2 x 106 cells) “ was replaced by “ (2 x 106 cells) “

Line 230 : “ method 32,33 “ was replaced by “ method [32,33] “

Line 316 : “ in treated cells is reported “ was replaced by “in treated cells are reported “

Line 336 : “ described 1“ was replaced by “described [1] “

Line 394 : “ AML cells 1,40 “ was replaced by “AML cells [1,40] “

Line 398 : “LXR-dependent gene expression “ was replaced by “LXR-dependent genes expression “

Round 2

Reviewer 1 Report

No comments.